# Impact of Feeder Access and Stocking Density on Tail Injuries and Performance in Weaned Piglets

**DOI:** 10.3390/ani15121749

**Published:** 2025-06-13

**Authors:** Anne Maria Stevina Huting, Francesc Molist, Piet van der Aar

**Affiliations:** Research & Development, Schothorst Feed Research B.V., 8218 NA Lelystad, The Netherlands; fmolist@schothorst.nl (F.M.); pietvanderaar@solcon.nl (P.v.d.A.)

**Keywords:** stocking density, pigs, faecal consistency, weaned piglets, feeders, performance, ear damage, behaviour

## Abstract

Tail injuries in pigs are a significant welfare concern in global pig husbandry. These injuries, often caused by tail biting, are influenced by several factors including feeder design, space allowances, and overall husbandry. Reducing tail-in-mouth behaviour from an early age is therefore essential. Inadequate housing conditions can lead to stress and abnormal behaviour, increasing the risk of tail injuries. This issue is particularly challenging when rearing pigs with intact tails, which requires higher welfare standards. This study showed that the welfare and/or performance of nursery pigs can be improved by ensuring proper feeder access and adequate space, thereby reducing the risk of tail injuries.

## 1. Introduction

One of the main welfare concerns in pig production systems is tail biting. To reduce the incidence of tail injuries, tail docking is commonly practised. However, tail docking does not address the underlying causes and is not always effective [1]. The EU directive 2008/120/EC states the following: ‘Neither tail-docking nor reduction of corner teeth must be carried out routinely but only where there is evidence that injuries to sows’ teats or to other pigs’ ears or tails have occurred. Before carrying out these procedures, other measures shall be taken to prevent tail-biting and other vices, taking into account environment and stocking densities. For this reason inadequate environmental conditions or management systems must be changed’. Nonetheless, in the Netherlands, tail docking will be prohibited from 2030 [2].

The biggest challenge with tail biting is that it is unpredictable and multifactorial, with various risk factors having been proposed [3,4]. One possible solution for reducing tail-directed behaviour is to lower stocking density, as limited space allowance can increase agonistic behaviour [5,6]. Currently, the minimum space allowance for weaned piglets (i.e., up to 30 kg body weight [BW]) according to EU Directive 2008/120/EC is 0.30 m^2^ per piglet. However, EU Directive 2010/63/EU, concerning the protection of animals used for scientific purposes, mandates a greater space allowance (i.e., 0.35 m^2^ per piglet between 10 and 20 kg and up to 0.50 m^2^ per piglet between 20 and 30 kg). Moreover, in the Netherlands, the aim is to increase the minimum space allowance for weaned piglets to 0.40 m^2^ per piglet by 2030 [2].

High stocking density is thought to affect pigs’ ability to thermoregulate, lying behaviour, disease transmission, competition for feed, and activity levels [1]. The space allowance per pig (in m^2^/pig·kg) is calculated using the following formula:*Space allowance/pig* = *k* × *BW*^2/3^(1)
where *k* is the space allowance coefficient, typically ranging from 0.03 to 0.05 [7], and *BW* is expressed in kg. The effectiveness of increasing space allowance alone, and its impact on agonistic behaviour, has not always been clear [8] and has mostly been studied in growing–finishing pigs [5,7,9,10,11,12,13,14,15]. However, tail-in-mouth behaviour begins at a young age and may predispose pigs to become biters later in life [16]. Therefore, early prevention of the onset of such behaviours is essential.

This study was performed to evaluate whether differences in stocking density and feeder access—within the bounds of current legislation—can influence the performance of weaned piglets and the prevalence of tail and ear injuries.

## 2. Materials and Methods

Three non-invasive experiments were conducted at the nursery unit of Schothorst Feed Research, with piglets housed in accordance with EU Directive 2008/120/EC.

### 2.1. General Protocol

Most management practices and measurement methods were consistent across experiments and are therefore described collectively in this section. Any deviations from these standard practices are detailed where applicable. Key information such as pen dimensions, feeder design, and floor type is discussed in Section 2.2, Section 2.3 and Section 2.4. Piglets (Large White × hybrid of Norsvin Landrace, Great Yorkshire, and Large White) were selected based on their body weight (BW) at weaning (day 0 [d0]), sex, and sow origin. Pre-weaning litters were not kept intact; instead, piglets were evenly distributed across the experimental treatments and mixed with unfamiliar individuals to form groups with uniform BWs. Only healthy piglets, with no signs of illness or injuries to the tail or ears, were included in the experiment.

#### 2.1.1. Pre- and Post-Weaning Management

Piglets were ear-tagged at birth, and no teeth clipping, tail docking, or castration was performed. Newborn piglets received an injectable iron supplement (1 mL, Iron-ject^®^; Dopharma, Raamsdonksveer, The Netherlands) at 3–4 days of age. They were not vaccinated either before or after weaning; however, the progenitors (gilts and sows) were vaccinated according to the manufacturer’s vaccination schedule with an inactivated vaccine against neonatal colibacillosis and *Clostridium* infections (SUISENG^®^ Coli/C; HIPRA, Amer Girona, Spain). From approximately 1 week of age until weaning, piglets were fed a commercial weaner 1 diet (CP 16%, SID Lys 10.1 g/kg, NE-swine 9.03 MJ/kg). The sow and her litter were housed in free farrowing/lactation pens measuring 2.50 m × 2.66 m.

The nursery pens were equipped with partially plastic slatted floors and solid plastic pen dividers. All rooms were windowless and climate-controlled. Room temperature was automatically regulated by a climate computer, following a temperature curve that started at 29 °C on the day of weaning (d0) and decreased to 22 °C by d35 post-weaning (PW). Ventilation was provided via a chimney in the weaning room, equipped with a fan and damper, and supplemented by an attic air inlet located in the corridor of the unit, allowing in fresh, cool outdoor air. Room humidity depended on both outdoor humidity and the ventilation rate. Artificial lighting (LED lights, 200 lux) was provided from 06:30 to 18:00 h.

The animals had ad libitum access to feed and water. The same commercial weaner 1 diet used pre-weaning was provided from d0 to d14 PW, followed by a commercial weaner 2 diet (CP 18%, SID Lys 11.6 g/kg, NE-swine 10.96 MJ/kg) from d14 to d35 PW. In experiment 3, the commercial weaner 2 diet had a CP content of 18.8%, an SID Lys content of 12.0 g/kg, and an NE-swine value of 10.9 MJ/kg. The dietary transition from weaner 1 to weaner 2 was abrupt.

The enrichment materials included a metal chain with a high-density polyethylene plastic bar, a metal chain with a solid plastic ball, a cotton rope, and a floor-based rubber toy.

#### 2.1.2. Health Checks and Medical Treatments

Piglets were inspected daily for general health. Any deviations and possible antibiotic treatments were recorded. Animals that reached the humane endpoint (see Appendix A for the definition and method of euthanasia) were humanely euthanised. In cases of mortality, the most likely cause of death was recorded.

#### 2.1.3. Measurements

A summary of the experimental conditions and measurements is presented in Table 1. All piglets were weighed individually on d0 (at a minimum weaning age of 28 days), d14 (a pen weight was used for experiment 3), and d35 PW. Average daily gain (ADG) was calculated on a pen basis for each phase (i.e., d0–14 and d14–35) and for the total period (i.e., d0–35). The coefficients of variation (CV, in %) for BWs at d0, d14 (except for experiment 3), and d35 PW were also calculated. Feed allowance and refusals were recorded per pen by weighing back the feed at the same time points as BW measurements. Average daily feed intake (ADFI) was calculated for each phase (d0–14 and d14–35) and for the total period (d0–35). The ADFI calculation accounted for piglets that were removed during the experiment by adjusting for the number of piglet-days (number of animals × number of days). Additionally, the feed conversion ratio (FCR) was calculated on a pen basis for each phase and for the total period.

Faecal consistency (FC) per pen was assessed twice a week using an 8-point scale, ranging from severe watery diarrhoea (score of 2) to hard, dry, and lumpy faeces (score of 9), with a score of 6 indicating normal FC, as described by Guan et al. [17]. Faecal scores were averaged for each experimental phase.

Piglets’ tails and ears were scored individually on d14 (except for experiment 1) and d35 PW using the classification system shown in Table 2. For experiments 2 and 3, an intervention protocol was implemented in the event of a tail- or ear-biting outbreak, as detailed in Appendix A.

### 2.2. Experiment 1: Stocking Density

Experiment 1 was conducted using a randomised complete block design with two experimental treatments: moderate stocking density (MSD) and low stocking density (LSD). Each treatment included 24 replicates (replicate = a pen consisting of a male/female ratio of 1:1 for all experiments). Stocking density was reduced by adjusting the group size (i.e., four or six piglets per pen).

For the MSD treatment, each replicate consisted of six piglets per pen (0.368 m^2^/piglet) (the stocking densities described in the current paper are the true space allowances after correcting for the feeder), housed in pens with a feeder containing three feeding spaces (2.00 piglets per feeding space) and one drinking nipple (push-lever bowl). For the LSD treatment, each replicate consisted of four piglets per pen (0.552 m^2^/piglet), with a feeder also including three feeding spaces (1.33 piglets per feeding space) and one drinking nipple (push-lever bowl). Pens, regardless of treatment, were equipped with partially slatted floors (27% solid floor) (see Figure 1 for more details).

The experiment was conducted across three successive weaning moments, with eight replicates per moment and 3 weeks between each. At all three weaning moments, a total of four experimental rooms per treatment were used, with two replicates per room.

**Figure 1 animals-15-01749-f001:**
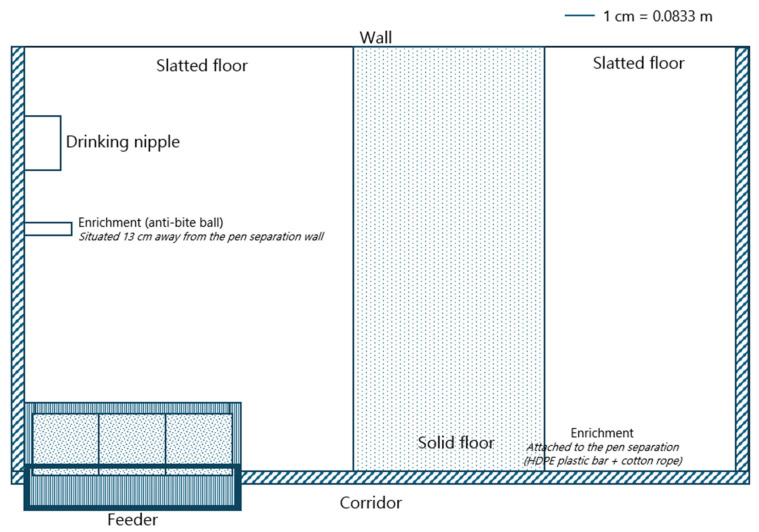
Illustration of nursery pens with partially slatted floors used in experiment 1. These pens were also used for the low feeder access (LFA1) group in experiment 2. The pen separation height was 0.85 m.

### 2.3. Experiment 2: Feeder Access 1

Experiment 2 was conducted as a completely randomised design with two experimental treatments: low feeder access (LFA1) and high feeder access (HFA1). Each treatment included 12 replicates. The LFA1 and HFA1 groups were housed in different experimental rooms.

For the HFA1 treatment, each replicate consisted of four piglets per pen (0.526 m^2^/piglet), with a feeder containing six feeding spaces (0.66 piglets per feeding space) and one drinking nipple (push-lever bowl). The pens for the HFA1 group were equipped with fully slatted floors (see Figure 2 for more details). For the LFA1 treatment, each replicate consisted of four piglets per pen (0.552 m^2^/piglet), with a feeder containing three feeding spaces (1.33 piglets per feeding space) and one drinking nipple (push-lever bowl). The pens for the LFA1 group were equipped with partially slatted floors (see Figure 1 for more details).

The experiment was performed over two consecutive weaning moments, with six replicates per moment and 3 weeks in between. At weaning moment 1, three experimental rooms per treatment were used, with two replicates per room. At weaning moment 2, one experimental room per treatment was used, with six replicates per room.

**Figure 2 animals-15-01749-f002:**
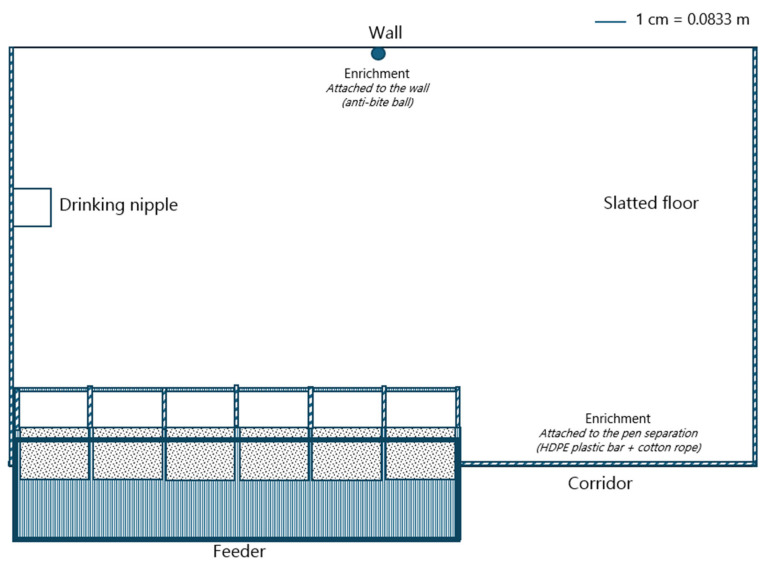
Illustration of nursery pens with fully slatted floors used in experiment 2. These pens were also used for the high feeder access (HFA2) group in experiment 3. The pen separation height was 0.78 m.

### 2.4. Experiment 3: Feeder Access 2

Experiment 3 was conducted using a randomised complete block design with two experimental treatments: low feeder access (LFA2) and high feeder access (HFA2). Each treatment included 12 replicates. The pens used for the LFA2 group were created by removing a pen separation wall (including the drinking nipple), as illustrated in Figure 3.

For the HFA2 treatment, each replicate consisted of six piglets per pen (0.350 m^2^/piglet), with a feeder containing six feeding spaces (1.00 piglets per feeding space) and one drinking nipple (push-lever bowl) (see Figure 2 for more details). For the LFA2 treatment, each replicate consisted of 14 piglets per pen (0.301 m^2^/piglet), with a feeder containing 6 feeding spaces (2.33 piglets per feeding space) and 1 drinking nipple (push-lever bowl). Pens for both HFA2 and LFA2 treatment groups were equipped with fully slatted floors.

The experiment was carried out at two weaning moments, with six replicates per moment and 6 weeks in between, using the same experimental rooms at each moment. At both weaning moments, two experimental rooms per treatment were used, with three replicates per room.

**Figure 3 animals-15-01749-f003:**
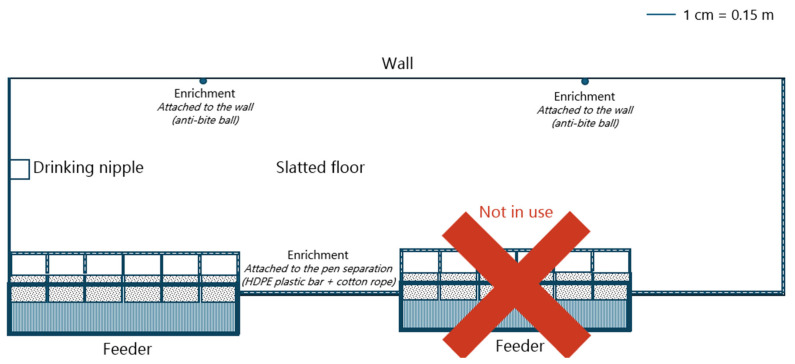
Illustration of the large nursery pens with fully slatted floors used in experiment 3. These pens were used for the low feeder access (LFA2) group. Only the left feeder was used to feed the piglets. The pen separation height was 0.78 m.

### 2.5. Statistical Analysis

The experimental data were analysed using GenStat^®^ version 23 for Windows™ (VSN International Ltd., Hemel Hempstead, UK). The Shapiro–Wilk test was used to assess the normal distribution of residuals, including residual plots (WSTATISTIC procedure), and Bartlett’s test was used to assess the homogeneity of variances (VHOMOGENEITY procedure). Observations of piglet performance (except for BW and CV at d0) were identified as outliers if the residual (fitted − observed value) exceeded 2.5 times the standard error of the residuals within the dataset. If at least one of the response parameters—ADFI, ADG, or FCR—was identified as an outlier, all three parameters for that experimental unit in that measurement period were treated as missing values. Missing values were estimated using least square estimates in GenStat^®^.

The pen was the experimental unit for ADFI, CV, ADG, BW, FCR, and FC. Performance data were analysed using one-way analysis of variance via the general linear model function, with experimental treatment as a fixed effect and weaning moment, room, and replicate as random effects for all experiments, as follows:*Y_ij_* = *µ* + *Block_i_* + *Treatment_j_* + *e_ij_*(2)
where *Y*_ij_ is the dependent variable; *µ* is the overall mean; *Block_i_* is the weaning moment, room, and replicate; *Treatment_j_* is the experimental treatment; and *e_ij_* is the residual error. A T-probability of *p* < 0.05 was considered statistically significant, while 0.05 < *p* ≤ 0.10 indicated a near-significant trend. Data are presented as mean and standard error of the mean (SEM) or standard deviation (Stdev), as applicable.

The experimental unit for tail and ear injuries was the number of piglets within a pen assigned to each injury score. Injuries were analysed using ordinal regression. Data were evaluated using a generalised linear model (GLM) with a multinomial distribution and logit-transformed data in GenStat^®^. Reported values represent the predicted percentage of each score within a treatment. The reported *p*-values are based solely on the GLM model. In this model, weaning moment, room, and replicate were included as a combined block.

## 3. Results

### 3.1. Experiment 1: Stocking Density

Piglets in experiment 1 were weaned at an average age of 29.9 days (Stdev = 1.13; range: 28–33 days), with an average weaning weight of 8.38 kg (Stdev = 0.535). In total, 13 of 240 piglets (5.42%) were medically treated. Treatments were administered for meningitis (six piglets), arthritis (five piglets), and respiratory problems (two piglets). Of the treated piglets, five belonged to the LSD group (5.21%), and eight to the MSD group (5.55%). One piglet (0.42%) died during the experiment. This piglet, from the MSD group, died of meningitis on d29 PW after having received medical treatment. Its final BW was 14.0 kg.

#### 3.1.1. PW Performance

Table 3 presents the effects of SD on piglet performance from weaning until d35 PW. SD (0.368 vs. 0.552 m^2^/piglet) significantly influenced ADG between d0 and 14, between d14 and 35, and across the full period (d0–35 PW). Piglets in the LSD group grew significantly more throughout the experiment compared with those in the MSD group, resulting in a tendency for higher BW at d35 PW (+900 g) in the LSD group. ADFI was significantly affected by SD between d14 and 35 and between d0 and 35 PW, with piglets in the LSD group showing higher intake than those in the MSD group. FCR between d0 and 14 PW was also significantly influenced, with LSD piglets having a lower FCR than MSD piglets. FC tended to be higher in the LSD group than in the MSD group between d0 and 14 PW.

#### 3.1.2. Damage Score

At d35 PW, 96.7% of piglets had intact tails, with 2.1% showing tail damage and 1.3% having an incomplete tail. Additionally, 99.6% of piglets had intact ears, while 0.4% had a damaged ear. Further details on the effects of weaning moment, room, and replicate on injury prevalence—as well as information on temperature differences between day and night—are provided in Appendix A.

SD did not significantly influence tail injuries at d35 PW (*p* = 0.14). Nonetheless, 97.9% of LSD piglets had intact tails (0.0% damaged, 2.1% incomplete), compared with 95.8% in the MSD group (3.5% damaged, 0.7% incomplete). In terms of ear injuries, only one piglet in the MSD group had a damaged ear at d35 PW (99.2% intact), while all piglets in the LSD group had intact ears (100%).

### 3.2. Experiment 2: Feeder Access 1

Piglets in experiment 2 were weaned at an average age of 31.4 days (Stdev = 1.66; range: 28–35 days), with an average weaning weight of 7.31 kg (Stdev = 0.383). In total, 4 of 96 piglets (4.17%) were medically treated. Treatments involved general weakness (three piglets) and arthritis (one piglet). Of the treated piglets, three belonged to the LFA1 group (6.25%) and one to the HFA1 group (2.08%). No piglets died during the course of the experiment.

#### 3.2.1. PW Performance

Table 4 shows the effects of FA1 on piglet performance from weaning until d35 PW. Feeder access 1 significantly influenced FC between d0 and 14 and between d0 and 35 PW and tended to influence FC between d14 and 35 PW. Piglets in the HFA1 treatment group had higher FC scores throughout the experiment than those in the LFA1 group. Additionally, FCR between d0 and 14 PW tended to be affected by FA1, with LFA1 piglets showing a higher FCR than HFA1 piglets. This was mainly due to the numerically lower ADG between d0 and 14 PW observed in the LFA1 group.

Figure 4 illustrates the effect of feeder access on FC throughout the experimental period. Consistent with the results presented in Table 4, piglets in the HFA1 group showed a lower average FC throughout the experiment, particularly during the first 2 weeks PW.

#### 3.2.2. Damage Score

At d14 PW, 99.0% of piglets had intact tails, with 1.0% showing tail damage, and 100% had intact ears. By the end of the experiment at d35 PW, 90.6% of piglets had intact tails, 6.8% had damaged tails, and 2.6% had incomplete tails. No ear damage was observed at d35 PW (i.e., 100% of piglets had intact ears). Further information regarding the effects of weaning moment, room, and replicate on the incidence of injuries; the temperature difference between day and night; and the use of the intervention protocol can be found in Appendix A.

Table 5 summarises the effects of FA1 on tail injuries at d14 and d35 PW. Piglets in the LFA1 group showed significantly more tail injuries at d14, but fewer tail injuries at d35 PW, compared with piglets in the HFA1 group.

### 3.3. Experiment 3: Feeder Access 2

Piglets in experiment 3 were weaned at an average age of 30.7 days (Stdev = 1.19; range: 28–34 days), with an average weaning weight of 9.21 kg (Stdev = 0.520). In total, 5 of 240 piglets (2.08%) were medically treated. Treatments involved arthritis (four piglets) and general weakness (one piglet), all of which occurred in the LFA2 group. Two piglets (0.83%) died during the experiment. One piglet from the LFA2 group died at d17 PW (final BW: 12.6 kg) due to intestinal complications, and one piglet from the HFA2 group died at d33 PW (final BW: 15.2 kg) due to gastric torsion.

#### 3.3.1. PW Performance

Table 6 presents the effects of FA2 on piglet performance from weaning until d35 PW. FA2 significantly affected only ADG and FCR between d14 and 35 PW. Piglets in the LFA2 treatment group had a significantly lower ADG (−41 g/day) and a significantly higher FCR (+0.05 g/kg BW) during this period compared with piglets in the HFA2 group. Consequently, LFA2 piglets were numerically lighter at d35 PW (−600 g) than their HFA2 counterparts.

#### 3.3.2. Damage Score

At d14 PW, 97.9% of piglets had intact tails, while 2.1% had damaged tails. Additionally, 99.6% had intact ears, with 0.4% showing ear damage. By d35 PW, only 45% of piglets had intact tails, 39.5% had damaged tails, and 15.5% had incomplete tails. No ear damage was observed at the end of the experiment (i.e., 100% of piglets had intact ears). Further information regarding the effects of weaning moment, room, and replicate on the incidence of injuries; the temperature difference between day and night; and the use of the intervention protocol can be found in Appendix A.

Table 7 illustrates the effect of FA2 on tail injuries at d14 and d35 PW. Piglets in the LFA2 group showed significantly more tail injuries at both time points compared with those in the HFA2 group.

## 4. Discussion

This paper presents three experiments investigating whether differences in SD and FA influence weaned piglets’ performance and agonistic behaviour. Variations in SD were achieved by altering group size (4 vs. 6 piglets per pen), while maintaining a constant feeder design with three feeding spaces per pen, regardless of the number of piglets. Differences in FA were introduced in two ways. In experiment 2, two facilities with similar SDs (Δ0.026 m^2^/piglet) were used: one with a six-space feeder in a fully slatted pen (HFA1), and the other with a three-space feeder in a partially slatted pen (LFA1). In Experiment 3, pen partitions were removed to increase the group size from 6 to 14 piglets, which altered both feeder access (1.00 vs. 2.33 piglets per feeding space) and SD (0.350 vs. 0.301 m^2^/piglet), forming the HFA2 and LFA2 groups, respectively. It was hypothesised that reduced SD would enhance piglet performance by minimising competition and disease risk, while increased FA would reduce agonistic behaviour, particularly under relatively high stocking conditions (±0.30 m^2^/piglet).

Housing systems in the current study were tested under experimental conditions with smaller group sizes than typically found in practice. For instance, weaned piglets are often housed in large groups during the PW period, after which they are split into smaller groups during the growing–finisher phase [18,19]. Conducting studies at a commercial scale, however, introduces challenges and risks related to confounding factors. O’Connell et al. [18] also suggested that group size alone does not affect pig performance if other conditions (e.g., space, density) are consistent, which supports the relevance of the current findings to practical settings. Furthermore, piglets were mixed in the PW period to create uniform BW groups and balance litters across treatments, helping to standardise early-life conditions and minimise pre-weaning effects [16,20] on later behaviour and performance.

High SD is known to impact several aspects of the housing environment—including air quality, ventilation, and pen hygiene—as well as pig behaviours, such as thermoregulation, lying, activity, and feeding competition, in addition to disease transmission risks [1,21,22,23]. High SD may also induce stress, which can influence metabolism, intestinal structure, and immune function [23]. However, the effects of space allowance on performance remain inconsistent across studies. This variability is partly due to differing definitions of what constitutes ‘adequate’ space, and the influence of confounding factors such as health status, group size, mixing practices, and pen design [24,25]. In the present study, increasing space allowance from 0.368 to 0.552 m^2^/piglet positively influenced PW performance. Piglets in the LSD group had a higher ADFI between d14 and 35 PW and overall, which contributed to an improved ADG. By d35 PW, LSD piglets were, on average, 0.9 kg heavier than those in the MSD group. This suggests that during the key growth period (d14–35 PW) [20], piglets may benefit from increased space and/or reduced feeder competition (1.33 piglets/feeder in LSD vs. 2.0 in MSD). These findings are consistent with those of Wolter et al. [26], who reported that performance may be more strongly influenced by feeder access than group size alone. Although ADFI did not differ notably during the first 14 days PW, LSD piglets showed a 9.4% higher ADG and tended to have improved FCR and FC. These improvements may reflect differences in feeding behaviour, particularly in the early PW period, when feeding is often socially facilitated [27,28]. Prior to weaning, piglets are accustomed to synchronised suckling, yet many commercial PW feeders do not support this behaviour. Still, some degree of feeder competition may be beneficial, helping to reduce feed waste and keep feed clean [29]. Mixing piglets can temporarily increase competition at the feeder until a new social hierarchy is formed [30]; thus, some competition is unavoidable. Chronic competition, however, can alter feeding patterns, increase feeding rate, or shift diurnal feeding rhythms [30,31,32]. Subordinate pigs may reduce their intake, often feeding at night, whereas dominant pigs tend to consume more feed during peak times [27,29,33]. Feeder availability was also evaluated in experiments 2 and 3. Pens with a piglet-to-feeder ratio of 0.66 (0.526 m^2^/piglet; HFA1) showed higher FC and tended to have better FCR in the early PW period compared with pens with a ratio of 1.33 (0.552 m^2^/piglet; LFA1). However, all FC scores remained within the normal range, and the differences were small. It is also possible that other variables—such as flooring or climate—had a stronger influence on performance than FA alone. In experiment 3, which used SDs permitted under commercial conditions, piglets with a piglet-to-feeder ratio of 2.33 (0.301 m^2^/piglet; LFA2) had lower ADG and higher FCR between d14 and 35 PW than piglets with a ratio of 1.00 (0.350 m^2^/piglet; HFA2). Increased competition for feeders may have contributed to this reduced performance, possibly due to the stress-induced metabolic impact during peak growth. Although ADFI was not significantly affected by FA in any of the experiments, feeding patterns—while not directly measured—may have been altered. Previous studies have shown that multi-space feeders can support synchronous feeding without affecting total feed intake or other performance parameters [27,31,32], as pigs might feed more effectively in competitive contexts [27]. No significant effects on BW CV were observed across the experiments, likely because of the relatively small group sizes used or low initial weight variability [34].

The reported incidence of tail injuries in pigs varies widely both between and within farms. When available, data are often based on slaughterhouse records [35], where prevalence rates of ≥35% are common [36,37]. In one of the few studies focusing on weaned piglets, injuries were found in 8% of ears and 4% of tails among Irish piglets [38]. These prevalences are comparable to the results from experiment 1 and the LFA1 treatment group in experiment 2 of the present study. On d35 PW (±25 kg BW), the proportion of piglets with intact tails was highest in experiment 1 (95.8%), followed by experiment 2 (90.6%), and lowest in experiment 3 (45.0%). Ear injuries were rarely observed across all experiments (<1%). The higher incidence of tail injuries in experiment 3 and the HFA1 group in experiment 2 may be partly attributed to housing on fully slatted floors, which has been identified as a risk factor for tail biting and other agonistic behaviours [39,40]. Additionally, tail biting is suggested to be more likely when the temperature difference between day and night is more extreme, a pattern common in autumn and spring [3]. Experiment 1 was conducted in the winter of 2021–2022, experiment 2 between autumn 2022 and winter 2023, and experiment 3 during the spring of 2024. Appendix A provide more details on temperature variation, which may help explain some differences in injury prevalence. However, the experimental factors themselves (i.e., SD, FA) may have played a more prominent role in explaining the differences observed between experiments.

Table 8 summarises studies evaluating the impact of SD on tail injuries in growing–finishing pigs, highlighting that the relationship between SD and agonistic behaviour remains inconclusive. Feeding motivation is one factor thought to influence the prevalence of tail biting [27,41], with competition for feed potentially increasing BW variation within groups. This variation can affect pigs’ ability to access feeders [42] and may increase the likelihood of forceful or sudden biting [43]. In experiment 1, SD influenced growth performance but did not significantly affect tail injuries, likely because of the low overall incidence [42]. By contrast, experiment 2 showed that piglets in the HFA1 group, despite better performance during the early PW period, had a higher prevalence of tail injuries by d35 PW than those in the LFA1 group. One potential explanation is that the HFA1 piglets were housed on fully slatted floors, a known risk factor for tail biting [39,40], despite being kept at relatively high space allowances (>0.520 m^2^/piglet). However, other environmental factors, such as ventilation, may also have contributed. Notably, the HFA1 and HFA2 groups exhibited similar levels of tail injuries at d35 PW, despite differences in space allowance (0.526 vs. 0.350 m^2^/piglet) and piglet-to-feeder ratios (0.66 vs. 1.00). Conversely, the combination of low space allowance (0.301 m^2^/piglet) and low feeder availability (2.33 piglets/feeder) in the LFA2 group was associated with an unexpectedly high incidence of tail injuries. Feeder access can strongly influence the prevalence of tail injuries, as previously reported [32,44], and this effect may be more pronounced in more vulnerable piglets, such as those housed at an SD of 0.30 m^2^/piglet, as seen in experiment 3. Additionally, experiment 3 introduced a substantial difference in group size (6 vs. 14 piglets per pen). Although some studies suggest that group size alone does not necessitate different space allowances [45], others indicate that larger group sizes may reduce performance and increase the risk of tail injuries [15,45], which may have influenced the outcomes observed in experiment 3.

## 5. Conclusions

Adjusting the stocking density from 0.37 to 0.55 m^2^/piglet (up to 25 kg BW) can positively influence piglet performance. Although stocking density was modified by altering group size—while maintaining the same feed allowance, as is common in practice—the feeder may have been the limiting factor rather than group size itself. A piglet-to-feeder ratio of ≤1.0 at a stocking density of 0.35–0.53 m^2^/piglet appeared to result in a similar incidence of tail injuries. However, in piglets that are more vulnerable due to a relatively high (albeit permitted) stocking density (0.30 m^2^/piglet) combined with high feeder competition (>2 piglets per feeder), a high incidence of tail injuries was observed. That said, the effectiveness of increasing feeder access may diminish when the overall prevalence of tail lesions is low (e.g., <5%).

## Figures and Tables

**Figure 4 animals-15-01749-f004:**
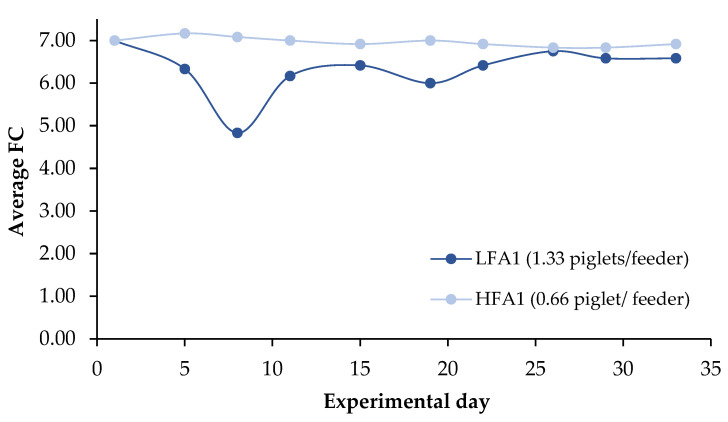
Effect of FA1 (LFA1 vs. HFA1) on FC throughout the experiment.

**Table 1 animals-15-01749-t001:** Summary of (**A**) experimental conditions and (**B**) measurements.

**A**	**Exp.**	**Trt**	**SD, m^2^/** **Piglet**	**FA, Piglets/** **Feeder**	**Piglets/** **Pen**	**Fig. Pen**	**Weaning Moments**
	1	Low SD (*n* = 24)	0.552	1.33	4	Figure 1	3 × 8 repl
		Moderate SD (*n* = 24)	0.368	2.00	6	Figure 1	
	2	Low FA1 (*n* = 12)	0.552	1.33	4	Figure 1	2 × 6 repl
		High FA1 (*n* = 12)	0.526	0.66	4	Figure 2	
	3	Low FA2 (*n* = 12)	0.301	2.33	14	Figure 3	2 × 6 repl
		High FA2 (*n* = 12)	0.350	1.00	6	Figure 2	
**B**	**Exp.**	**Comments**	**Body Weight, kg**			**FC**	**Tail and Ear Injuries**
	1	Trt within same room, winter 2021/2022	d0 (indiv.), d14 (indiv.), and d35 (indiv.) PW			2×	d35 PW
	2	Trt within different rooms, autumn 2022– winter 2023	d0 (indiv.), d14 (indiv.), and d35 (indiv.) PW			2×	d14 and d35 PW
	3	Trt within same room, spring 2024	d0 (indiv.), d14 (pen), and d35 (indiv.) PW			2×	d14 and d35 PW

SD = stocking density, FA = feeder access, Exp. = experiment, Trt = experimental treatment, Fig. = figure illustrating pen design, *n* = number of replicates, repl = replicate, indiv. = individual, PW = post-weaning, and FC = faecal consistency × weekly.

**Table 2 animals-15-01749-t002:** Classification of tail and ear injuries.

Tail injuries		
Score 0	Intact tail	No tail damage
Score 1	Damaged tail	Bite marks, open wounds (fresh blood or crust), and/or red/swollen tails
Score 2	Incomplete tail	Shortened tail
Ear injuries		
Score 0	Intact ear	No ear damage
Score 1	Damaged ear	Superficial scratches or visible injuries with fresh or dried blood
Score 2	Incomplete ear	Severe damage, part of ear is missing

**Table 3 animals-15-01749-t003:** Effects of SD (MSD vs. LSD) on PW piglet performance and FC.

Parameter	LSD 1.33 Piglets/Feeder 0.552 m^2^/Piglet 4 Piglets/Pen	MSD 2.00 Piglets/Feeder 0.368 m^2^/Piglet 6 Piglets/Pen	SEM	*p*-Value
BW, kg				
d0	8.38	8.38	0.005	0.61
d14	11.6	11.4	0.11	0.14
d35	24.7	23.8	0.33	0.06
CV, %				
d0	1.46	1.28	0.106	0.25
d14	7.72	7.53	0.595	0.83
d35	10.2	11.9	1.01	0.24
ADG, g/day				
d0–14	255 ^b^	231 ^a^	7.9	0.04
d14–35	598 ^b^	564 ^a^	9.4	0.02
d0–35	471 ^b^	442 ^a^	9.6	0.05
ADFI, d/day				
d0–14	316	306	9.9	0.48
d14–35	846 ^b^	781 ^a^	14.0	0.01
d0–35	667 ^b^	617 ^a^	9.3	0.01
FCR, g/kg BW				
d0–14	1.24 ^a^	1.33 ^b^	0.022	0.02
d14–35	1.41	1.39	0.015	0.28
d0–35	1.43	1.40	0.023	0.38
FC				
d0–14	6.67	6.45	0.090	0.09
d14–35	6.90	6.95	0.062	0.63
d0–35	6.81	6.76	0.048	0.40

Note: Different superscripts (^ab^) within a column indicate a significant difference between treatments (*p* < 0.05).

**Table 4 animals-15-01749-t004:** Effects of FA1 (LFA1 vs. HFA1) on PW piglet performance and FC.

Parameter	LFA1 1.33 Piglets/Feeder 0.552 m^2^/Piglet 4 Piglets/Pen	HFA1 0.66 Piglet/Feeder 0.526 m^2^/Piglet 4 Piglets/Pen	SEM	*p*-Value
BW, kg				
d0	7.31	7.31	0.148	0.99
d14	9.95	10.4	0.455	0.56
d35	21.9	22.6	0.58	0.39
CV, %				
d0	1.25	1.16	0.183	0.75
d14	10.1	7.70	1.067	0.18
d35	11.0	11.2	1.72	0.95
ADG, g/day				
d0–14	183	218	26.7	0.40
d14–35	563	584	16.9	0.42
d0–35	411	438	15.3	0.28
ADFI, d/day				
d0–14	303	297	27.2	0.89
d14–35	773	767	33.5	0.90
d0–35	585	579	29.1	0.89
FCR, g/kg BW				
d0–14	1.71	1.38	0.094	0.06
d14–35	1.39	1.32	0.058	0.39
d0–35	1.43	1.33	0.045	0.16
FC				
d0–14	6.08 ^a^	7.06 ^b^	0.137	0.01
d14–35	6.46	6.90	0.142	0.08
d0–35	6.31 ^a^	6.97 ^b^	0.124	0.02

Note: Different superscripts (^ab^) within a column indicate a significant difference between treatments (*p* < 0.05).

**Table 5 animals-15-01749-t005:** Effect of FA1 (LFA1 vs. HFA1) on tail injuries at d14 and d35 PW.

	LFA1 1.33 Piglets/Feeder 0.552 m^2^/Piglet 4 Piglets/Pen	HFA1 0.66 Piglet/Feeder 0.526 m^2^/Piglet 4 Piglets/Pen	*p*-Value
d14			
Intact, %	95.8	100.0	0.01
Damaged, %	4.2	0.0	
Incomplete, %	0.0	0.0	
d35			
Intact, %	93.9	87.1	0.01
Damaged, %	5.6	11.3	
Incomplete, %	0.5	1.7	

**Table 6 animals-15-01749-t006:** Effects of FA2 (LFA2 vs. HFA2) on PW piglet performance and FC.

Parameter	LFA2 2.33 Piglets/Feeder 0.301 m^2^/Piglet 14 Piglets/Pen	HFA2 1.00 Piglet/Feeder 0.350 m^2^/Piglet 6 Piglets/Pen	SEM	*p*-Value
BW, kg				
d0	9.21	9.21	0.001	0.49
d14	12.5	12.4	0.09	0.89
d35	25.2	25.8	0.25	0.25
CV, %				
d0	0.774	0.774	0.0144	1.00
d35	10.7	9.49	0.976	0.39
ADG, g/day				
d0–14	227	226	6.1	0.91
d14–35	608 ^a^	649 ^b^	10.8	0.03
d0–35	456	473	7.2	0.13
ADFI, d/day				
d0–14	299	319	11.3	0.24
d14–35	848	870	16.3	0.37
d0–35	625	634	14.5	0.68
FCR, g/kg BW				
d0–14	1.34	1.47	0.070	0.19
d14–35	1.40 ^b^	1.35 ^a^	0.015	0.04
d0–35	1.37	1.34	0.020	0.26
FC				
d0–14	6.69	6.71	0.060	0.77
d14–35	6.83	6.78	0.065	0.60
d0–35	6.72	6.73	0.048	0.90

Note: Different superscripts (^ab^) within a column indicate a significant difference between treatments (*p* < 0.05).

**Table 7 animals-15-01749-t007:** Effect of FA2 (LFA2 vs. HFA2) on tail injuries at d14 and d35 PW.

	LFA2 2.33 Piglets/Feeder 0.301 m^2^/Piglet 14 Piglets/Pen	HFA2 1.00 Piglet/Feeder 0.350 m^2^/Piglet 6 Piglets/Pen	*p*-Value
d14			
Intact, %	97.0	100.0	<0.001
Damaged, %	3.0	0.0	
Incomplete, %	0.0	0.0	
d35			
Intact, %	28.5	85.3	<0.001
Damaged, %	50.5	13.3	
Incomplete, %	21.0	1.4	

**Table 8 animals-15-01749-t008:** Overview of the literature on effects of SD on tail injuries.

Reference	BW (kg)	m^2^/Pig	Estimated *k* Final BW	Applied by Adjusting	Slatted Floor	Space Feeders, *n*	Tail Injuries by Lowering SD
Beattie et al. [5]	15–45	0.5–1.3	0.040–0.103	Pen size	Partially	4	No difference
Brandt et al. [9]	20–110	0.70 vs. 0.89	0.030–0.039	Pen and group size ^1^	Partially	Trough	Fewer injuries
Cornale et al. [10]	25–110	1.00–1.50	0.044–0.065	Group size	Partially	Multi	No difference
Ewbank and Bryant [11]	20–60	0.56–1.19	0.036–0.078	Pen size	Partially	4	Fewer agonistic interactions
Jensen et al. [12]	30–90	1.00 vs. 0.64	0.032–0.049	Group size	Partially	Tube feeder	No difference
Klaaborg et al. [13]	30–110	0.77–1.00	0.034–0.044	Pen and group size ^1^	Partially	Trough	No difference
Larsen et al. [14]	30–110	0.73–1.21	0.032–0.052	Group size	Partially	3	No difference
Street and Gonyou [15]	35–100	0.52–0.78	0.023–0.036	Pen and group size	Fully	1	No difference
Vermeer et al. [7]	30–90	1.2, 1.6, 2.4	0.059–0.120	Pen size ^2^	Partially	6 or 12	Fewer injuries

Note: *k* (space allowance coefficient) was estimated on the basis of the final BW of the pig. ^1^ The pen design was completely different between treatments, which included differences in area in solid flooring, straw allocation, space allowance, and group size. ^2^ By removing pen separations.

## Data Availability

The data presented in this study are available on request from the corresponding author. The data are not publicly available due to their proprietary nature.

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
