# Peer review of "Impact of Feeder Access and Stocking Density on Tail Injuries and Performance in Weaned Piglets"

_animals, 2025, doi:10.3390/ani15121749_

Round 1

Reviewer 1 Report

Comments and Suggestions for Authors

This paper addresses an important topic, as tail biting is a common welfare issue across various stages of pig production. The manuscript is generally well written. However, I recommend that it be reviewed by a proficient English speaker to improve clarity and flow. I support the publication of this manuscript, but only after a few revisions—particularly in the Methodology and Discussion sections. The Methodology should provide clearer details regarding dates, descriptions, and potential confounding factors. Additionally, the Discussion should more thoroughly acknowledge confounding factors and it may need to include a few lines addressing piglet mixing after weaning, as this is highly relevant to the parameters measured in this study. Please, review the points below.

  1. Abstract: It would be interesting to know here how many piglets were studied in total if you have space.
  2. L23: FCR or FC? If feed conversion rate, please write in full the first time.
  3. L25: ADG needs to be written out in full the first time it appears
  4. L50: If weight is given in kg and space allowance is given in m2/pig, the unit of the k coefficients (0.03 to 0.05) has to be m2/(pig⸱kg) and should be stated. If you want to use the International System of Units, your units could alternatively be stated as in the example: m2⸱pig-1, m2⸱(pig⸱kg)-1, but because your units have the word “pig” in them, I do not think that using the “/” sign should be of any problem.
  5. L70-91: Please, add a statement so that the reader knows that there are more details about the pen in section 3.2, ex “(please see section 3.2 for more details). Otherwise, it takes a few lines to realize that important information such as the floor opening (27%), dimensions etc, is indeed described in the paper. Please, state the pen’s height in section 3.2 and in Figure 1’s legend. I also suggest adding some information about where are these piglets coming from in terms of housing (farrowing pens, crates, outdoor huts, and main enclosure and management characteristics).
  6. L81-83: Please add room dimensions and number/size/type of ventilation fans and inlets (cooling pads? Attic inlets? Tunnel doors?).
  7. L83: Could you please state the type of lighting, just so the reader knows approximately the colour and intensity range?
  8. L87 and elsewhere: standardize, either use “experiment 3” (my suggestion), or Trial 3 as in the abstract.
  9. L94: please consider adding the description of the human endpoint as supplementary material.
  10. L95: Please, state the method of euthanasia instead of “humane”, which is subjective.
  11. L105-107: Please, state clearly when/how was the feed weighted (once at the commencement and end of each period or regularly? If the later, how often?).
  12. L112: Typo at “and”.
  13. L120: Did you have 24 replicates or Experimental Units, EUs? How many piglets in total? If 24 replicates, please state how many replicates of 6 piglets (12?) and how many with 4 piglets (12?).
  14. L121: please state the reference “2” outside of the “()”, otherwise it seems that you are stating area per piglet^2.
  15. L123 and elsewhere: “nipple drinker” or “drinking nipple” instead of “drink nipple”
  16. Lines 117-127: What did you block for? Room, weaning moments? How many rooms (2)? Within a room you had two replicates, where these of distinct treatments or same treatments? Please, clarify in the text or add a reference for where the information is.
  17. L131-164: Please, add the applicable clarifications added to Experiment 1.
  18. L176: “Missing values were estimated through GenStat.” can possibly be removed, as you have the same statement on L179.
  19. Line 185 (model): If you are blocking replicate, were your blocks “room” and “weaning moment” really necessary statistically speaking? Otherwise, wouldn’t you be just wasting degrees of freedom in your model? If all these blocks were statistically necessary, please state so in the results section and/or add this information to the supplementary material.
  20. L197: replace “moment” for “weaning moment”.
  21. L98-100: Please, add some context of why FC is a relevant parameter to measure with a citation.
  22. Table 2: Please check to see if the journal requires also the X- (or F-value, or the applicable value) and DFs in addition to the p-value.
  23. L235 and elsewhere: Should HSD be the MSD treatment?
  24. L239 and elsewhere: Please, add ± std for weaning age for all experiments. Make weaning age for all experiments clear in the Material and Methods section as well.
  25. L269: typo “in”.
  26. Table 4: needs units of measurement.
  27. Discussion L321: If feeder size was not adjusted in pens with 6 and 4 piglets, your SD results are somewhat confounded with FA and this needs to be better acknowledged. Also, were replicates of 6 and 4 piglets from the same sow? This is highly relevant as negative interactions will be more frequently when piglets are mixed. How was the process of selecting littermates for each replicate in the nursery pens?
  28. In Methodology: Please, state the FA values for your SD groups. LSD and MSD had different FA ratios, and this needs to be clear in the methodology as well.
  29. L412: Add date/season information of each weaning moment to the methodology section.
  30. K423: correct “didn’t had”.
  31. I suggest that this paper is reviewed by an English native/proficient speaker for a 
    more polished writing. 
Comments on the Quality of English Language

31. I suggest that this paper is reviewed by an English native/proficient speaker for a more polished writing.

Reviewer 2 Report

Comments and Suggestions for Authors

This article compiles the results of three experimental studies. It investigates the impact of stocking density and feeder access on performance and damaging behaviours.

As far as I’m concerned, there are two major limitations in this article:

First, the Materials and Methods (M&M) section is difficult to read, as the experiments are sometimes mixed together, and there is a lack of coherence in the writing. A table summarizing the experimental conditions would be more useful than the current illustration of the pens. Please ensure a systematic and consistent description of the experimental conditions for each study.

Second, I find it confusing that the conditions were not consistent between the compared groups. In particular, the floor type (fully vs. partially slatted) and group size differed, which could have influenced feeding and damaging behaviours. In my opinion, this issue is insufficiently discussed in the article.

I will read the discussion section more thoroughly once these two major issues, as well as the minor points detailed below, are addressed.

Detailed Comments:

  • L10 – “It is thereby key to already decrease …”: The phrase is grammatically awkward.
  • L34-35 – The statement is vague. It would be more precise and informative to cite the specific EU Directive.
  • L36 – Please provide the full reference for the cited text.
  • L42 – You cite “EU Directive 2008/120/EC” here but not on lines 34–35. Please ensure consistency in how references are cited throughout the manuscript.
  • L65 – Tempo x TN70 are not genetic lines in the strict sense (e.g., Pietrain, Large White). Please revise accordingly.
  • L75 – Suiseng® alone is not the name of a vaccine. Specify the full commercial name, e.g., Suiseng® Coli AC or Suiseng® Diff A.
  • L83 – Use consistent formatting for time: “06:30 until 18:00”.
  • L90 – Please define HDPE at first mention.
  • L189 – A ‘<’ symbol is missing. Please correct.
  • L203 – SD stands for stocking density in your article. Clarify this in line 203. Also, in line 190–191 you wrote “Data are presented as means and SEM” — please ensure consistency across the results section.
  • L204 – “Medical treatments”
  • L207–201 – Simplify the sentence: “One piglet from the MSD group died of meningitis at D29 PW.”
  • L222 – Rephrase to: “The effect of stocking density (moderate [SDM] or low [LSD]) …”
  • L242 – Again, “medical treatment”.
  • L239–267 – Although there is a statistically significant difference in fecal scores, the actual difference (less than one point except on D10) may be minor in practical terms. For example, FC = 6 vs. FC = 7 may not lead to different management decisions. Consider whether this discussion is relevant and necessary.
  • L377–383 – See the comment above. Can such a minor difference in fecal scores truly support the conclusion that the pigs were healthier?
  • L304–316 – I am not an expert in behavioural studies, but the group size difference (6 pigs in HFA2 vs. 14 pigs in LFA2) should be more thoroughly discussed regarding its potential influence on damaging behaviour.
  • Table 5 & 6 – Please indicate the number of pigs per pen.
  • Tables 3 & 4 – Please specify the space per pig in m² (m²/pig).
